# Interview study exploring how global health partnership principles are enacted and recommendations for practice

Rebecca Rose Turner [ID],[1] Jo Hart [ID],[1] Natalie Carr,[1] Eleanor Bull,[1] Jessica Fraser,[1,2] Lucie Byrne-Davis [ID] [1]

¹Health Workforce Group, Division of Medical Education, The University of Manchester, Manchester, UK
²Tropical Health and Education Trust (THET), London, UK

**Correspondence to**
Dr Lucie Byrne-Davis;
lucie.byrne-davis@manchester.ac.uk

## ABSTRACT

**Background** Effective global health partnerships can strengthen and improve health and healthcare systems across the world; however, establishing and maintaining effective partnerships can be challenging. Principles of Partnerships have been developed to improve the quality and effectiveness of health partnerships. It is unclear how principles are enacted in practice, and current research has not always included the voices of low-income and middle-income country partners. This study aimed to explore how The Tropical Health and Education Trust's nine Principles of Partnership are enacted in practice, from the points of view of partners from low-income, middle-income and high-income countries, to help improve partnerships' quality and sustainability.

**Methods** People who had been a part of previous and/or ongoing health partnerships were interviewed virtually. Participants were purposefully sampled and interviews were conducted using an appreciative inquiry approach. Audio recordings were transcribed and deductive framework analysis was conducted.

**Results** 13 participants from 8 partnerships were interviewed. Six participants were based in the low-income or middle-income countries and seven in the UK. Key findings identified strategies that enacted 'successful' and 'effective' partnerships within the Principles of Partnerships. These included practical techniques such as hiring a project manager, managing expectations and openly sharing information about the team's expertise and aspirations. Other strategies included the importance of consulting behavioural science to ensure the partnerships consider longevity and sustainability of the partnership.

**Discussion** Core principles to effective partnerships do not work in isolation of each other; they are intertwined and are complimentary to support equitable partnerships. Good communication and relationships built on trust which allow all partners to contribute equally throughout the project are core foundations for sustainable partnerships. Recommendations for established and future partnerships include embedding behavioural scientists/psychologists to support change to improve the quality and sustainability of health partnerships.

## STRENGTHS AND LIMITATIONS OF THIS STUDY

⇒ The participants of this study were all participating in a single scheme focused on antimicrobial stewardship, how they enacted Principles of Partnership might be different to other participants with different project goals.

⇒ We took a deductive approach to exploring partnership principles, building on previous work in this area.

⇒ We used appreciative inquiry to structure the interviews; this has the benefit of allowing participants to explore challenges without focusing on the negative, but might also prevent participants from raising failures.

## BACKGROUND

Health partnerships are a way to improve health and healthcare across the world via the exchange of knowledge and skills, building capacity and codeveloping interventions between individuals and organisations from different countries.[1] Global health partnerships can be defined as long-term established collaborations between high-income countries (HICs) such as the UK, and low-to-middle-income countries (LMICs) that work together to achieve a common goal.[2] The long-term aims of health partnerships vary greatly and previous examples have addressed antibiotic use in healthcare settings,[3] helped prioritise mental well-being in communities, as well as enabling educating and training of workforces to further build capacity.[4 5]

Establishing successful health partnerships is a complex intervention in itself. Whilst they often work towards a common aim, establishing a successful partnership where teams work effectively, collaboratively and respectfully is challenging. Global health partnerships encompass a complex intersection of individual motivations, different professional groups, diverse funding opportunities and

vast health and social inequities that continue within and between countries across the world.[6] A growing body of literature has started to explore the key determinants to understanding what makes a health partnership successful or not. A recent evidence synthesis highlighted commonalities within global health partnerships.[6] Authors identified professional relationships as important for partnerships; however, many failed to address equity within their partnerships. For example, some did not allow all partners to actively contribute to discussions, highlighting power dynamics still exist within global health partnerships.[7] Another challenge within health partnerships is the issue of colonialism. This is due to stakeholders from HICs driving projects based on their own priorities, leading them to impart their 'superior' Western knowledge on to health workers in LMICs.[8] This can lead to a hierarchy, with HICs often referred to as the 'donors', and the hosting country known as the 'recipient'.[9]

Organisations such as the Tropical Health and Education Trust (THET) and the WHO are the fundamental organisations in global health partnerships. THET manages UK aid funding to support access to healthcare across the world.[1] Examples of these include the 'Commonwealth Partnerships for Antimicrobial Stewardship' (CwPAMS) scheme, where health partnerships aim to address antimicrobial resistance challenges in LMICs and in the UK through bidirectional exchange.[10] THET have developed key principles for health partnerships to improve their quality and effectiveness. These are in the form of nine Principles of Partnership (see table 1).[11] One study used the THET eight Principles of Partnership (the ninth principle has since been added) as a framework to review a global partnership between the Ministry of Health in Nepal, a healthcare provider and University in California.[12] Key findings included a need for efficient working practices to minimise the cost of integration and an attempt to prioritise the national and host institution's goals while concurrently endeavouring to develop the careers of global health workers.

Further research is required to explore how THET's Principles of Partnership, including the new ninth principle (*Embed equity and inclusion*), are enacted within health partnerships from both LMIC and HIC partners. There is a lack of practical and equitable understanding and support for future healthcare workers and researchers of how partnerships can be established and maintained. It is also unclear what exactly makes these partnerships work successfully together, if there is evidence about which of these principles are specifically important for change and how they are enacted. By gaining further understanding about what makes health partnerships work well in practice, we can replicate key elements in future partnerships to support effective working and provide practical examples.

The principles of health partnerships as developed by THET[7] were explored in CwPAMS partnerships as part of this study, focusing on these health partnerships within

**Table 1** Definitions of the THET nine partnership principles

| Principles | Definition as provided by THET |
|---|---|
| Harmonised and aligned with national plans | Health partnerships' work is consistent with local and national plans and complements the activities of other development partners. |
| Effective and sustainable | Health partnerships operate in a way that delivers high-quality projects that meet targets and achieves long term results. |
| Strategic | Health partnerships have a shared vision, have long-term aims and measurable plans for achieving them and work within a jointly agreed framework of priorities and direction. |
| Respectful and reciprocal | Health partnerships listen to one another and plan, implement and learn together. |
| Organised and accountable | Health partnerships are well structured, well managed and efficient and have clear and transparent decision-making processes. |
| Responsible and build trust with partners | Health partnerships conduct their activities with integrity and cultivate trust in their interactions with stakeholders. |
| Flexible, resourceful and innovative | Health partnerships proactively adapt and respond to altered circumstances and embrace change. |
| Committed to joint learning | Health partnerships monitor, evaluate and reflect on their activities and results, articulate lessons learned and share knowledge with others. (The THET. THET: Principles of Partnership (available from: https://www.thet.org/principles-of-partnership/) |
| Embed equity and inclusion | Health Partnerships consider unequal power relations and inequalities experienced by individuals as a result of their social identities and conduct GESI activities and analysis to ensure GESI is mainstreamed into organisations, programmes, interventions and activities. |

GESI, Gender Equality and Social Inclusion; THET, Tropic Health and Education Trust.

one clinical area will act as an exemplar focus. While there are many key organisations within global health, who promote strategies to encourage successful partnerships, THET is one of the major organisations in health partnership work in the UK. The aim of this study was to explore and understand how the Principles of Partnerships are enacted within health partnerships, to centre the voices of LMIC partners and to provide practical recommendations for current and future partnerships.

## METHODS

### Study design

This was a qualitative, semi-structured interview study, guided by criteria for reporting qualitative research Consolidated criteria for Reporting Qualitative research,[13] included in online supplemental material. The topic guide explored how the key partnership principles advocated by THET[7] were enacted. Appreciative Inquiry (AI)[14] was adopted during the design of the interview questions and by the interviewers throughout the interviews. An AI approach was used to encourage people to talk explicitly about what worked well, what strengths people displayed and how they imagine a successful partnership could be further established.[15] After data analysis, recommendations were made to help improve partnerships' quality and sustainability. These recommendations were developed by the research team, based on professional judgement and experience in developing theory-based recommendations for policy and practice.

### Research team

RRT is a woman and health psychology researcher with an MSc and PhD in health psychology, working as a postdoctoral research associate. JH is a woman, professor of health professional education, an Health and Care Professions Council (HCPC)-registered health psychologist, with an MSc and PhD in health psychology. NC is a woman with an MSc in health psychology, currently undertaking a PhD in international health workforce. EB is a woman, an HCPC-registered health psychologist, with an MSc and PhD in health psychology, working in public health and the NHS. JF is a woman working as programmes coordinator in an international non-governmental organisation (INGO). She has a Bachelor's degree in international social policy and a masters in global health. LB-D is a woman, a professor of health psychology and an HCPC-registered health psychologist, with an MSc and PhD in health psychology. EB was embedded in one of the health partnerships and LB-D and JH supervised behavioural science volunteers in five of the other health partnerships.

### Participants

This context was selected as the partnerships were all based within the same clinical area (antimicrobial stewardship) and a handful of geographical areas. Due to the specific aim of the study, studying a relatively homogenous group of health partnerships within one clinical area and using a specific framework to focus the study, purposive sampling was used.[16] Participants consisted of individuals who were involved, or had previously been involved in a global health partnerships exploring antibiotic use in Commonwealth countries. Participants were not paid for participation in this study.

### Recruitment

An initial recruitment email was sent to all potential participants from a representative at THET, asking interested participants to express an interest in the study. There were 12 partnerships but the final number of potential participants is unknown to us, as the partnerships themselves vary in terms of how many people are involved at any one time in the partnerships. We estimate that there would be at least four participants per partnership, giving us a minimum potential pool of participants of 48. Potentially interested participants agreed for their contact details to be passed onto a member of the research team. An email was then sent to these participants with a participant information sheet outlining the purpose of the study, what procedures would take place, data protection and confidentiality, as well as relevant contact information. Willing volunteers responded to the email and interview dates were organised. Informed consent was gained from participants at the start of each interview and recorded, with the interviewer reading out each statement on the consent form, and participants providing their consent. We included all willing participants.

### Data collection

Interviews were conducted over Zoom or MS Teams by RRT or NC, who had no prior relationship to the interviewees. Interviewees were encouraged to take part in a confidential space. Interviews followed a semistructured interview schedule (included in online supplemental materials) as well as probed and follow-up questions based on participant's answers. Interviews lasted approximately 45–75 mins. Interviews were recorded and transcribed using the autotranscription tool via Zoom or MS Teams. Transcriptions were checked for accuracy, adjusted to intelligent verbatim transcriptions and anonymised by redacting any information about people, places or organisations. Transcripts were then uploaded into NVIVO 12 V.12.5.0.815. Field notes were not used. Sample size was determined by the specific narrow aim of the study and homogeneity of the group[16] and in line with pragmatically with recruitment constraints.

### Data analysis

Data were analysed using a deductive framework approach,[17] using the nine Principles of Partnerships developed by THET as the framework.[7] RRT initially familiarised themselves with the data by listening to the audio tapes and re-reading the transcripts. Any key ideas or thoughts were listed at this stage. The framework was applied to each of the transcript using NVIVO 12 V.12.5.0.815, coding each significant piece of test into the framework. Where text related to more than one principle, it was coded to both. Once all the transcripts were coded, LB-D, JH and EB checked 100% of the data between them and checked their significance in relation to the particular framework domain. Where there were differences, these were discussed and resolved between that coder and RRT. Data was then summarised by RRT. RRT then continued with the analysis taking a thematic approach, generating themes, identifying differences and similarities between participants from the HICs and LMICs and connections between the principles. At this

| Table 2 | Characteristics of participants (n=13) | |
|---|---|---|
| Profession | Pharmacist | 8 |
| | Microbiologist | 1 |
| | Project manager | 2 |
| | Academic | 2 |
| Country | Kenya | 2 |
| | Ghana | 2 |
| | Malawi | 2 |
| | UK | 7 |

point, quotes were selected to demonstrate the breadth and depth of the theme within the framework.

## RESULTS
### Characteristics of participants
A total of 13 individuals from 8 partnerships took part in an interview, the majority were pharmacists (69%) and other participants were academics (15%), microbiologists (8%) and project managers (8%). Six participants were based in the LMIC partner (Kenya, Ghana or Malawi) and seven participants were based in the HIC partner (UK). Due to the partnerships having a relatively small number of people and our commitment to anonymity, we have included characteristics that are unlikely to deanonymise the participants (ie, profession and country) only. Please see table 2. Participants are identified after each quote by an assigned number, for example, P1.

### Principles of Partnerships
Findings are presented narratively under each partnership principle.

### Harmonised and aligned with national plans
To ensure the health partnerships were aligned with national plans, the teams typically carried out three actions (1) consulted national policy documents, (2) met with local governments and policy-makers and (3) adapted guidance from the UK for use in the LMIC.

National policy documents were consulted at the beginning of the partnership and throughout to ensure the interventions that were developed were in line with the national agendas.

> So right from the beginning, we had copies from the [LMIC] team of their national guidance of their national Antimicrobial Resistance protocol … we were all tasked to read them to make sure that what we were devising and doing fitted in quite clearly with the governmental approach that the [LMIC team] was taking. (P9: Pharmacist, UK)

Participants from HIC discussed how they would work with local (LMIC) teams to adapt the principles of UK guidelines, which often acted as a starting point for teams. However, it was crucial that individuals from the LMIC local context supported the translation of these guidelines into local policies.

> There may be some general principles that are applicable wherever you're working, but you gotta adapt those locally and the people who are gonna help us do that are the people who work closely. (P1: Microbiologist, UK)

Participants from LMICs stated the importance of building on this by holding regular meetings with key stakeholders within healthcare to further ensure connectivity to national plans and establish relationships with senior people within the healthcare systems.

> We have been involved in different meetings with the Minister of Health. We meet routinely once a quarter or sometimes a month, depending on the need and discuss Antimicrobial Stewardship issues … That helps us with our partnership aligned to the national strategy in the country. (P11: Pharmacist, Malawi)

### Effective and sustainable
Sustaining health partnerships for the long-term was a key priority, not only to continue to build capacity in LMICs but also to keep partnership workers engaged. While it was acknowledged this cannot be guaranteed, organisations and senior leaders were seen as having a key role to play in supporting partners in the long-term.

> Nothing is fundamentally ever sustainable, but I think the key elements are, we've got fantastic support now from the hospital leadership. The whole hospital has become aware of a significant change on the ward. (P5: Academic, UK)

Creating this much needed support from health partnerships organisations and leaders was established by showcasing the benefits that volunteering brings to individuals such as new skills, which are then translated into their everyday practice.

> I think the institutional support we have from our trust is fantastic and I don't think we could do that without it. So that's certainly a key factor and I think just giving us that in believing in us. I think the institution and organization sees the value of, for example, workforce development, it brings innovation, it brings those kind of benefits that volunteers come back with. (P6: Pharmacist, UK)

Ensuring the partnership was effective and sustainable also led to the use of behavioural science, while individuals in the partnership did not profess to being experts in this field, they found drawing on the expertise of applied psychologists and behavioural scientists valuable. Drawing on this field enlightens the partnerships teams and supported them in thinking about long-term implementation and behaviour change, often something they described as not previously doing.

The whole psychologists and influencing behaviour changes was a game changer for me. I think we're trying to bring in some of those principles with our own antibiotic interventions we're trying to develop locally. I think that recognizing how fundamental that is and how overlooked it is within many aspects for me and my colleagues working in [redacted] …I think that if we can get people from The Change Exchange [behavioural science volunteering programme] embedded in all the partnerships that would make it very big difference into the way things are delivered. (P1: Microbiologist, UK)

Participants from LMICs described how it was helpful for colleagues trained in behaviour change to deliver this aspect, as participants understood behaviours needed to change but did not have the expertise to support this.

You needed to change certain behaviors. So our colleagues introduced the COM-B model, you know, we try to get to what was going on and then after trainings we conducted another survey to see whether there has been any improvement since. Obviously your colleagues from The Change Exchange led the behavioural change aspect because they were the specialists within behaviour change. That was easy. (P14, Pharmacist, Ghana)

### Strategic

Having a long-term vision for the health partnership was discussed among partnerships, so while the projects within the partnership are of a smaller scale, they were always working towards their overall vision and aim. Both HIC and LMIC participants discussed the constant desire to be considering the next step and exploring where the partnership could go.

So it's kind of having that big vision and recognizing that even though what we're doing seems very small scale like you know guideline writing, developing committees that create guidelines, it's all part of that bigger vision. So it's maintaining that and sharing that bigger vision. (P1: Microbiologist, UK)

We're always talking. We had a WhatsApp group. We're always discussing, even after the meetings when they had gone, you know, to their guest house and all that, we're still on WhatsApp. We're still talking how we can improve on what we are doing (P12: Project manager, Ghana)

Ensuring health workers were supported within the team to continue to pursue other projects within the overall aim of the partnership was important in maintaining an effective and collegiate team, where individual's skills could develop.

I try to mobilize my colleagues, we need to do more proposals to support the project to run, because what is most important is that there should be activities for people to do, if there are not activities … People will

lose vision. So, once they lose the vision, they will scatter. (P3: Pharmacist, Kenya)

### Respectful and reciprocal

Having a historical relationship between the partners and everyone being familiar with each other in the partnership was attributed to success. This also helped the experience to be enjoyable and participants saw their colleagues as friends. The formation of friendships helped to maintain respect among the partners.

I think firstly, it's because we already have that strong partnership. So there's a natural and good sort of healthy working relationship there anyway. We don't just class them as colleagues, we class them as friends. (P6: Pharmacist, UK)

For new partnerships where professional and personal relationships had not necessarily developed yet, respect and mutual trust developed by setting tasks completed by both teams. This perception that people are doing what they said they would, helped to build up a foundation of trust, as being reliable and accountable was viewed as integral for the partnership from both HIC and LMIC participants.

You know, we make meetings and they attend when they say they're going to, we do. So there's a certain amount of trust that you build up that way. (P4: Pharmacist, UK)

Participants from HIC discussed how LMIC found it 'refreshing' that HIC colleagues showed respect to colleagues in LMICs by not being forceful in their approach and thinking they know best.

There was a comment from somebody from the one of the medical personnel in [redacted] that we met almost saying, oh, it's quite refreshing that you're not coming over here and telling us what we should be doing and we're like, well, no, that's not what we're here for, you know. (P13, Pharmacist, UK)

### Organised and accountable

Having individuals or small teams who were specifically responsible for ensuring the partnership was organised and kept to specific timelines, such as project managers, were described by both HIC and LMIC participants as vital for ensuring the partnership was delivered in an effective and timely manner. Partnership teams noticed the difference in performance when project managers were appointed in helping to deliver the project and tasks.

[Project manager] was instrumental in organizing the Zoom meetings, and she has worked all round to ensure everything is going on well in the project. (P2: Pharmacist, Kenya)

I think we're very lucky with in [redacted] that we've got support from Global Health Partners, which are full time people who work to support parts of activity

happening … They've been absolutely fabulous and trying to ensure that we stick the timetables. (P1: Microbiologist, UK)

LMIC participants viewed it as important that project managers/steering committees were balanced and equal.

So we have what we call the steering committee within our partnership … the committee was also based was also constituted visually equal equality. So you've got a balance across board. (P14: Pharmacist, Ghana)

HIC participants discussed developing a work plan that was agreed by all, at the beginning of the specific project. This was useful tool to keep the partnership organised and accountable. This work plan also clearly highlighted expertise, roles and expectations, so the professional relationships and workload could be managed.

We have a document with them which upholds exactly what we set out to achieve and we refresh that and we make sure we review that sort of time and time again, we'll probably be doing some again now post COVID. It definitely I think it shows a firm commitment and not just on a personal level, but actually an institutional commitment, which I think is really helpful in terms of getting the support and resources that you need behind the partnership. (P6: Pharmacist, UK)

### Responsible and build trust with partners

Building trust within partnerships was described as something that 'took time' and something that occurred due to familiarity and development of friendships. Having a face-to-face connection and being able to spend time outside of work with colleagues helped to further develop trust that was described by HIC participants.

But having had that time together, you know not just in work, but also in a social context as well … you've got that kind of friendship element as well to it, which I think in the longer run is you know what we need in the partnership that sort of camaraderie and banter that comes with having had that sort of face-to-face interaction I think is important. (P13: Pharmacist, UK)

Whereas colleagues from LMICs talk about building trust from transparency and keeping each other accountable. They describe trust building each time a task has been completed from both LMIC and HIC colleagues.

For the whole project to be successive, each and everyone else have to do their part, so it created that mutual respect where no one feels like they're the ones doing more than the other party … So over time and seeing the progress that we have been making, I think that helped us to devote that trust and that mutual respect. (P11: Pharmacist, Malawai)

### Flexible, resourceful and innovative

Allowing the partnership to be flexible, especially during the COVID-19 pandemic, was important to allow everyone to contribute equally. Often things would change during the course of the partnership and participants discussed having to think innovatively about how to overcome certain obstacles.

We had to have some contingency measures like due to one of them, or an example was when maybe flights were not possible to be used. We have some contingency plans that we had, whereby one of them was the Zoom meetings. (P2, Pharmacist, Kenya)

Yeah, and thinking innovatively and I think being flexible as well because it's not always time differences and stuff like that. It's not always easy as it seems, but you will make it happen. (P6: Pharmacist, UK)

Regular communication with partners and funders was expected, especially if changes needed to be made due to unexpected delays. This helped the partners continue to work towards their overall aim but within the boundaries of the partnership that were could often change quickly. Participants also discussed the importance of meeting regularly as a team, so no one had to make difficult decisions on their own.

So when things come up, which we have to modify things, we notify each other as partners since we already in good contact then we also notify the Commonwealth Pharmacy Association and they were necessary it's because we got especially we need is to do with the change of our expenditure of money and also things that to do with the change of the protocols or change of the implementation plan of the project because we cannot just hold the goals. (P11: Pharmacist, Malawi)

### Committed to joint learning

A core element was considered to be the continued learning that takes place between both countries. This often involved sharing experiences, tools and supporting each other with other issues outside of the specific aims of the partnership.

The real key is that what it says the partnership, it's not about doing it one way, it's about it's about learning both ways. I think that we've learned so much from my [redacted] team as well. It's very much not being a one way thing. (P1: Microbiologist, UK)

Learning was often perceived as a consequence of the partnerships, but also encouraged throughout by sharing materials or mentoring people. This often went beyond the boundaries of the health partnerships to support people with their careers or preparation for similar issues, such as the COVID-19 pandemic.

I suppose, behind what we're doing, you know, is trying to say we're here to support, you know, because we've done this ourselves locally, it's trying to support you in doing the same thing, you know, in your context. (P13: Pharmacist, UK)

### Embed equity and inclusion

Ensuring the partnership foundations were based on equality, diversity and inclusion was considered a fundamental aspect to establishing the health partnership. This was sometimes formally monitored, with formal documents developed to ensure equality. These documents were signed when joining the health partnership, with consequences if breached. These ideas and formalisation of ideas often came from the LMIC participants.

> I think still based on the grounds of equality and diversity, we ensure, that's there's mutual respect and not just ensuring this. We've got documents that's back, whatever we do and there are sanctions for anyone that goes against the sort of offense, another party, so they, are laid down procedures to ensure that things are streamlined and properly formalized. Every member literally signed onto it. So I mean you've consented to abide by the rules, you go against it the loss would be with you. (P14: Pharmacist, Ghana)

The focus and fundamental principle of equality appeared to drive flexibility within the partnership, by allowing the ability to be flexible and considerate of others.

> There's been that flexibility that has really helped, so that everyone is able to contribute equally do their tasks without yeah burdening one person or burdening few people. (P8: Project Manager, Malawi)

A way in which the partnership ensured people were able to contribute equally in a practical sense was by using different platforms to communicate and edit documents.

> I think, using a platform like Google Drive has helped us, and the weekly meetings have helped us to be able to share information equally and to share tasks equally. So I think that's basically how we've been doing it. (P8: Project Manager, Malawi)

### Recommendations for practice

To ensure this research promotes further effective practice in health partnerships, recommendations in line with the Principles of Partnerships have been developed, these are presented in table 3.

### DISCUSSION

This study identified how key principles of health partnerships are enacted within partnerships, specifically within CwPAMS, as an exemplar. Key findings of this study highlight the importance of good communication and transparency, the role of behavioural scientists/psychologists in ensuring sustainability and allowing all involved in the health partnership to have the opportunity to be heard and contribute fully.

This study supports previous findings, such as difficulties in communication, which include language barriers,[18] or variations in language and behaviours among cultures[19] can be barriers to successful global health partnerships. Similar to findings by Jenkins *et al* (2021), who found relationships that continue to flourish outside of work where friendships form, and shared appropriate language that invokes 'camaraderie' or 'banter' is fundamental in creating trust and transparency when building partnerships.[20] While it may be important to monitor the frequency, type and effectiveness of communications in order to evaluate partnership relationships,[21] this would not take into account communication and relationships that occur outside of work in a social context. However, a communication plan detailing when and how meetings or updates will take place will help to reduce miscommunication between partners, aligning to both the *responsible* and the *respectful and reciprocal* Principles of Partnerships.

Behavioural scientists and psychologists were identified as playing a crucial role in the *Sustainability and effectiveness* of health partnerships. Previous research has identified the benefits of behavioural science within health partnerships[4 22] and carious projects between the UK and Uganda, Sierra Leone and Mozambique have demonstrated how behavioural scientists can help to strengthen global health partnerships while improving local health systems.[4 5] The Change Exchange (behavioural science volunteer programme) has aimed to bridge this gap by placing behavioural scientists in health partnerships including those in this study.[4] As the partnerships all rely on behaviour change, whether that's behaviour change of clinical teams or communities to be successful, adding in experts in this area can help improve the effectiveness of the partnerships.

The COVID-19 pandemic has highlighted the importance of the *flexible, resourceful* and *innovative* principle. Partners have been forced to adapt to an ever-changing pandemic, therefore demonstrating openness and flexibility are fundamental to flourishing partnerships. Furthermore, the COVID-19 pandemic disrupted in-person activities, while exacerbating existing inequalities in global health, especially in LMICs.[23 24] Teams had to quickly adapt and rely on virtual interactions and approaches.[2] Research suggests there are differences between what LMIC and HIC partners value in virtual aspects of partnerships[2] and differences in how trust is established virtually compared with face to face.[25] Having face-to-face contact has been associated with increased initial trust, compared with virtual, but over time, levels of trust are comparable between virtual and face-to-face teams.[25] Establishing trust was viewed as a central component of partnerships. Being able to have that human interaction, coupled with each team completing tasks, was found to increase trust.

Technology and shared drives have enabled individuals to contribute equally to discussions and documents. However, poor internet connections and conflicting daily priorities among institutions can cause difficulties in arranging meetings and document sharing between LMICs and HICs.[2 26] Thus, while individuals appear *committed to joint learning*, even during a global pandemic,

**Table 3** Recommendations for practice to support building a successful partnership

| Principle | Example of enactment | Recommendation |
|---|---|---|
| Harmonised and aligned with national plans | Whatever we do, we try to consult the national policy documents in the national action plan so that at the end of the day, our actions are aligned … So you know there's always interconnectivity between whatever we are doing. (P14: Pharmacist, Ghana) | Guidance from the high-income country, if relevant, should always be adapted for local use. Engaging with local stakeholders around how such guidance may work within the local context is good practice. |
| Effective and sustainable | There was a big Behavioural change, two of them on the team that came and so they helped with the activities … They shared with us, some modules and some metrics that we could use as we needed to continue with the project and actually if we were able to get the staff to accept it and to understand. (P12: Project manager, Ghana) | Experts in behavioural science if embedded in health partnerships multidisciplinary teams can help to address behaviour change and long-term change. These experts can work on interventions and evaluations to drive and understand change. |
| Strategic | So it's maintaining that and sharing that bigger vision. (P1: Microbiologist, UK) | Health partnerships should focus on developing a long-term, overarching vision and strategic aim(s). While partnerships may focus on small projects at a time, it is recommended that the health partnerships are formed as a partnership, rather than a project team. Showcasing the work within the partnerships can help to raise awareness of their benefits. Using case studies, writing up papers for publication and disseminating information is recommended. |
| Respectful and reciprocal | The relationships were being built on kind of basic based on kind of mutual respect and friendship really. (P1: Microbiologist, UK) | When setting up health partnerships, it is important to recruit individuals with the same values and vision. Carrying out informal interviews to understand their experience and values is recommended. |
| Organised and accountable | The massive gear change was halfway through the project. We had the project manager appointed. (P13: Pharmacist, UK) | Appointing a project manager and/or steering committee is recommended at the beginning of the project. Regular communication via platforms such as shared drives or messenger platforms which are acceptable for the team is recommended. This communication should be set out in a communication plan, so everyone is aware of when communication is expected but also how this communication with happen. |
| Responsible and build trust with partners | I think it's been built up over years. I wouldn't say it was just, you know, job done on day one kind of thing. I definitely think that's been cultivated over years and years. (P6: Pharmacist, UK) | While participants discussed that health partnerships can work successfully without face-to-face visits, these are recommended where appropriate, as this can help to build rapport and trust between teams. Face-to-face visits also help teams gain a further understanding of the context, which can help to better develop interventions. |
| Flexible, resourceful and innovative | There's been that flexibility that has really helped, so that everyone is able to contribute equally do their tasks without burdening one person or burdening few people. (P8: Project Manager, Malawi) | Developing contracts to state expectations of how individuals should treat others and be treated, can help to eliminate any problems in ways of working and set boundaries within the partnership. Developing a work-plan with agreed time plan that allows for flexibility is recommended. This should be co-developed and regularly reviewed as a team. |
| Committed to joint learning | It's kind of this partnership of what we can share and the so much we've learned so much they've learned. (P9: Pharmacist, UK) | Including a bidirectional learning pathway in funding and timelines, where both teams share and learn from each other, could include visits/placements in both countries. Sharing CVs at the start of the project can help individuals identify mentors or expertise for further professional development. |
| Embed equity and inclusion | We do have a strategy which is gender equity or inclusion which we've decided to and incorporated into the process. (P2: Pharmacist, Kenya) | Health partnership embedding equity and inclusion is essential. Formalising this in the form of a strategy is recommended as this can ensure people are treated equally and supported appropriately as individuals. People can hold others accountable if such strategies are breached. It also helps to set expectations for people who may join. |

CV, Curriculum Vitae.

it is important that the partnership considers the technological capacity within the partnership in order to maintain *respect* and *reciprocity* for all members of the partnership.[2]

National governments in LMICs often change priorities for their own countries to meet the demands of international partners with their own agenda.[27] For participants within this study, following local and national plans ensured partnerships were *harmonised* and *aligned*. Arguably, this gives individuals direction, and will ensure they are working towards the goal of the project without compromising their own governments' guidelines. It was acknowledged that UK guidelines cannot be implemented into LMICs without change; they must be altered to the local community while also using the skills of behavioural scientists and psychologists.

Critically, many studies have failed to include voices of all partners involved in global health partnerships, suggesting that, while many authors discuss equity, power relations are still prevalent.[21 28] Additionally, critical reflection about implementation to support the Principles of Partnership do not feature in much published work.[29] This study attempted to overcome the issue of lack of representation by recruiting participants from both HICs and LMICs. However, this research is not without its limitations, interviews were carried out using an AI approach[14] and while this is a strength and allows the research team to explore common strengths in establishing and working in health partnerships, the barriers to enacting principles were less apparent. Further research should explore how behaviours within partnerships are enacted and what the influences on these behaviours are, to make a partnership 'work'. Further, the power relations are such a crucial part of partnerships that a robust exploration of power and its influence on enactment of partnership principles requires more attention. As identified above, partners critically reflecting on their partnerships is something that is important but often not studied. In this study, we did not specifically ask the partners whether they themselves spent time reflecting on the partnership and the enactment of the principles. A study of this reflection in and on partnership practice is warranted if we are understanding more deeply about what drives the enactment (or not) of principles. Another limitation is the interviews were conducted online due to participants being in different parts of the world. Carrying out the interviews online, while practically allowed for the research to be carried out, come with limitations as this can impact on people's ability to be able to take part in the study due to requiring internet access and the appropriate devices. We did not present the findings and associated recommendations back to the participants or partnerships. This is a limitation, as it does not allow for grounding and checking with the participants, and should be a focus of future work.

## CONCLUSION

Health partnerships are complex, with many different interacting parts. The nine Principles of Partnership developed by THET outline key elements to establish and develop a 'successful' partnership that goes beyond a partnership achieving its short-term project goals. Ten practical recommendations are suggested to support future health partnerships in enacting these principles, which include expectation setting from the inception of health partnerships, the role of behavioural scientists/psychologists in supporting health partnerships and continuing to highlight the importance of their work. Further work should continue to explore how to establish, sustained and evaluate health partnerships.

**Acknowledgements** We would like to thank the study participants for their time and engagement in the study.

**Contributors** LB-D and JH acquired the funding for this study. LB-D, JH, EB and RRT conceived the study design. RRT and NC collected the data, with recruitment support from JF. RRT led the qualitative analysis, with support from LB-D, JH and EB. All authors revised the first manuscript draft and read and approved the final manuscript. LBD is the overall guarantor of the manuscript.

**Funding** This project was part of the Commonwealth Partnerships for Antimicrobial Stewardship (CwPAMS), managed by the Tropical Health and Education Trust (THET) and Commonwealth Pharmacists Association (CPA). CwPAMS is funded by the UK Department of Health and Social Care's Fleming Fund using UK aid. The views expressed in this publication are those of the authors and not necessarily those of the UK Department of Health and Social Care, the NHS, the represented NHS Trusts, CPA or THET.

**Competing interests** JF works for the Tropical Health and Education Trust. LB-D, JH and EB have received funding from THET to conduct this research and other studies. No other conflicts of interest to declare.

**Patient and public involvement** Patients and/or the public were not involved in the design, or conduct, or reporting, or dissemination plans of this research.

**Patient consent for publication** Not applicable.

**Ethics approval** This study involves human participants. This study has received ethical approval from the University of Manchester University Research Ethics Committee 5 (Ref: 2021-12956-21469). Participants gave informed consent to participate in the study before taking part.

**Provenance and peer review** Not commissioned; externally peer reviewed.

**Data availability statement** Data are available upon reasonable request. Data are from a small number of participants from a small population. We will provide data if we can be assured of the anonymity of our participants, commensurate with the undertaking in our research ethics approval.

**ORCID iDs**
Rebecca Rose Turner http://orcid.org/0000-0002-0480-4626
Jo Hart http://orcid.org/0000-0001-9985-5137
Lucie Byrne-Davis http://orcid.org/0000-0002-9658-5394

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
