## [Reviewer comments · BMJ Open]

ARTICLE DETAILS

TITLE (PROVISIONAL)	An interview study exploring how global health partnership principles are enacted and recommendations for practice
AUTHORS	Turner, Rebecca; Hart, Jo; Carr, Natalie; Bull, Eleanor; Fraser, Jessica; Byrne-Davis, Lucie

VERSION 1 – REVIEW

REVIEWER	Baillie, Jodie The University of Sydney, University Centre for Rural Health
REVIEW RETURNED	08-Aug-2023

GENERAL COMMENTS	This study aims to explore how principles of partnerships are implemented in practice. Here are some suggestions for refinement: 1. It is advised to curtail the usage of acronyms, employing them solely when the pertinent organization or terminology recurs three or more times within the text.2. The rationale for your paper would be strengthened by noting that critical reflection to support the implementation principles is largely absent from published literature. Baillie J, Laycock AF, Conte KP, et al. Principles guiding ethical research in a collaboration to strengthen Indigenous primary healthcare in Australia: learning from experience. BMJ Global Health 2021;6:e003852.3. You oscillate between the terms 'enact principles' and 'implement principles'. Enact and implement have similar but different meanings. Please be clear on terminology.4. Participants - How many participants were invited to interview.5. Participants - Did you look to get a coverage of people from different countries, organisation types, roles? Please expand on the criteria for sampling.6. Recruitment – were participants paid to participate in interviews? Please state.7. Data analysis – 'The principles of partnerships were used as a framework using a coding manual.' This is a repeat of above where you say 'Data was analysed using a deductive framework approach (16), using the nine principles of partnerships developed by THET as a framework' – consider streamlining – saying once – or differentiating.8. Data analysis – did LBH, JH and EB individually check all the data? How did you resolve differences in interpretation? This needs to be further explained and articulated.9. Data analysis – when you say 'LBH, JH, and EB checked 100% of the data' does this mean you checked the output provided by RT or did you individually co-code the data?10. Data analysis – Who did the analysis on the differences and similarities between participants and connections. What was the
--

	process for this to occur? This needs to be detailed so readers can assess the methodological rigour and understand process. 11. Data analysis – ‘Following the framework analysis ...,’ maybe you need to put this into the study design? 12. Data analysis – Did you take preliminary findings back to the partnership? Please add this in to manuscript. 13. Data analysis – as you mention the research teams role in developing the recommendations it would be useful to add an authors note saying who you are and your role and experience. This is outlined as good practice in COREQ guidelines and will help with understanding who is doing this work. I am curious as you why you did not take the recommendations back to the partnership to workshop. 14. Methods – How did you select the quotes displayed in the results? At what stage? Who did this? 15. Results – Table 2. Please remove the percentages. The numbers are too small to express as percentage. 16. Results – Characteristics that should also be included (based on your stated inclusion criteria) relate to existing or previous member of the partnership, how long people were involved in the partnership. Were these people working on the same partnerships? 17. Results – first quote there seems to be a typo in the participant description. 18. Discussion – missing was any real reflection on the role of power in these partnerships. 19. Discussion – more reflection on the process of reflecting on principles would have really added to this manuscript. How did this process work? In work we have done we found that people found it difficult to think about principles individually. 20. Limitations – draft findings were not presented back to the partnership in a broader sense to ‘ground – truth’ and check. 21. Limitations – the research team are embedded within the partnership? Major limitation as you potentially have a vested interest to show in a positive light. Thanks for the opportunity to comment.
--	---

REVIEWER	Alharbi, Khulud The University of Manchester Faculty of Medical and Human Sciences
REVIEW RETURNED	23-Oct-2023

GENERAL COMMENTS	Abstract: The study's title mentioned challenges, but the aim was to explore implementation and the result highlighted strategies for effective partnership. It appears that there is a mismatch between the aim, title, and result. Result My suggestion is to include a table comparing the two parties, showing factors that support or hinder implementation.
---

VERSION 1 – AUTHOR RESPONSE

Reviewer: 1 Ms. Jodie Bailie, The University of Sydney, The University of Sydney	
---	--

Comments to the Author: This study aims to explore how principles of partnerships are implemented in practice. Here are some suggestions for refinement:	
1. It is advised to curtail the usage of acronyms, employing them solely when the pertinent organization or terminology recurs three or more times within the text.	Thank you – we have retained acronyms where they are used multiple times eg LMIC and AMR, and replaced others with the full words.
2. The rationale for your paper would be strengthened by noting that critical reflection to support the implementation principles is largely absent from published literature. Bailie J, Laycock AF, Conte KP, et al. Principles guiding ethical research in a collaboration to strengthen Indigenous primary healthcare in Australia: learning from experience. BMJ Global Health 2021;6:e003852.	Thank you for the suggestion and the interesting reference – we have added that in as part of the discussion.
3. You oscillate between the terms ‘enact principles’ and ‘implement principles’. Enact and implement have similar but different meanings. Please be clear on terminology.	Thank you – we have checked through usage of each throughout – we mean enact rather than how people are implementing.
4. Participants - How many participants were invited to interview.	We have explained this further – 12 partnerships were invited, and partnerships vary in size.
5. Participants - Did you look to get a coverage of people from different countries, organisation types, roles? Please expand on the criteria for sampling.	As above, recruitment was through 12 existing partnerships in 5 countries. We included all willing interviewees.
6. Recruitment – were participants paid to participate in interviews? Please state.	They were not paid and we have clarified this.
7. Data analysis – ‘The principles of partnerships were used as a framework using a coding manual.’ This is a repeat of above where you say ‘ Data was analysed using a deductive framework approach (16), using the nine principles of partnerships developed by THET as a framework’ – consider streamlining – saying once – or differentiating.	Thank you for spotting this – we have reworded and streamlined
8. Data analysis – did LBH, JH and EB individually check all the data? How did you resolve differences in interpretation? This needs to be further explained and articulated.	We have further clarified and explained how we resolved differences.
9. Data analysis – when you say ‘LBH, JH, and EB checked 100% of the data’ does this mean you checked the output provided by RT or did you individually co-code the data?	We checked the output and checked their significance to the domains – we have clarified this in the text.

10. Data analysis – Who did the analysis on the differences and similarities between participants and connections. What was the process for this to occur? This needs to be detailed so readers can assess the methodological rigour and understand process.	We have added some detail on this point.
11. Data analysis – ‘Following the framework analysis ...,’ maybe you need to put this into the study design?	We have now added this in
12. Data analysis – Did you take preliminary findings back to the partnership? Please add this in to manuscript.	We didn’t do this. It would be good to do this for future work
13. Data analysis – as you mention the research teams role in developing the recommendations it would be useful to add an authors note saying who you are and your role and experience. This is outlined as good practice in COREQ guidelines and will help with understanding who is doing this work. I am curious as you why you did not take the recommendations back to the partnership to workshop.	We have added in details about the team; into a section in methods. This was a small scale study and we wouldn’t have been able to add in that workshop stage. We will be interested in feedback and discussion about the recommendations with people who read the paper. We are continuing to work with the partnerships on a new phase of their funded work.
14. Methods – How did you select the quotes displayed in the results? At what stage? Who did this?	We have added more detail into data analysis.
15. Results – Table 2. Please remove the percentages. The numbers are too small to express as percentage.	We have removed the percentages from Table 2.
16. Results – Characteristics that should also be included (based on your stated inclusion criteria) relate to existing or previous member of the partnership, how long people were involved in the partnership. Were these people working on the same partnerships?	The partnerships are small with great variability within them – it is not possible to anonymously include this information. We have added this clarification in.
17. Results – first quote there seems to be a typo in the participant description.	Thanks – we have modified that
18. Discussion – missing was any real reflection on the role of power in these partnerships.	We understand this is crucially important. We are not experts in this area and have now suggested this as an area for future research.
19. Discussion – more reflection on the process of reflecting on principles would have really added to this manuscript. How did this process work? In work we have done we found that people found it difficult to think about principles individually.	We had added some suggestions into the discussion about how these could be considered further.

20. Limitations – draft findings were not presented back to the partnership in a broader sense to ‘ground – truth’ and check.	We have added this as a limitation
21. Limitations – the research team are embedded within the partnership? Major limitation as you potentially have a vested interest to show in a positive light. Thanks for the opportunity to comment.	We have explained this as part of the research team section so that people are able to see our working relationships with the partnerships. We have clarified a potential conflict of interest.
Reviewer: 2 Mrs. Khulud Alharbi, The University of Manchester Faculty of Medical and Human Sciences Comments to the Author: Abstract: The study's title mentioned challenges, but the aim was to explore implementation and the result highlighted strategies for effective partnership. It appears that there is a mismatch between the aim, title, and result.	Thanks for highlighting this, we have removed mention of challenges in the title.
Result My suggestion is to include a table comparing the two parties, showing factors that support or hinder implementation.	As we didn't explore challenges in depth, we don't feel able to be able to construct this table – we have changed the title to reflect this.